# The Bacterial Genomic Context of Highly Trimethoprim-Resistant DfrB Dihydrofolate Reductases Highlights an Emerging Threat to Public Health

**DOI:** 10.3390/antibiotics10040433

**Published:** 2021-04-13

**Authors:** Claudèle Lemay-St-Denis, Sarah-Slim Diwan, Joelle N. Pelletier

**Affiliations:** 1Department of Biochemistry and Molecular Medecine, Université de Montréal, Montréal, QC H3T 1J4, Canada; claudele.lemay-st-denis@umontreal.ca (C.L.-S.-D.); sarah.slim.diwan@umontreal.ca (S.-S.D.); 2PROTEO, The Québec Network for Research on Protein, Function, Engineering and Applications, Québec, QC G1V 0A6, Canada; 3CGCC, Center in Green Chemistry and Catalysis, Montréal, QC H3A 0B8, Canada; 4Chemistry Department, Université de Montréal, Montréal, QC H2V 0B3, Canada

**Keywords:** antibiotic resistance, resistome, type II dihydrofolate reductase, genomic context, mobile genetic elements

## Abstract

Type B dihydrofolate reductase (*dfrb*) genes were identified following the introduction of trimethoprim in the 1960s. Although they intrinsically confer resistance to trimethoprim (TMP) that is orders of magnitude greater than through other mechanisms, the distribution and prevalence of these short (237 bp) genes is unknown. Indeed, this knowledge has been hampered by systematic biases in search methodologies. Here, we investigate the genomic context of *dfrbs* to gain information on their current distribution in bacterial genomes. Upon searching publicly available databases, we identified 61 sequences containing *dfrbs* within an analyzable genomic context. The majority (70%) of those sequences also harbor virulence genes and 97% of the *dfrbs* are found near a mobile genetic element, representing a potential risk for antibiotic resistance genes. We further identified and confirmed the TMP-resistant phenotype of two new members of the family, *dfrb10* and *dfrb11*. *Dfrbs* are found both in *Betaproteobacteria* and *Gammaproteobacteria*, a majority (59%) being in *Pseudomonas aeruginosa*. Previously labelled as strictly plasmid-borne, we found 69% of *dfrbs* in the chromosome of pathogenic bacteria. Our results demonstrate that the intrinsically TMP-resistant *dfrbs* are a potential emerging threat to public health and justify closer surveillance of these genes.

## 1. Introduction

Trimethoprim (TMP) is a synthetic antimicrobial that is ranked as being highly important by the World Health Organization (WHO) [1]. TMP strongly and selectively inhibits a key enzyme in bacterial folate biosynthesis, chromosomal dihydrofolate reductase (K_I_ (*Escherichia coli* FolA) = 20 pM [2]), thereby effectively abolishing bacterial proliferation. This antimicrobial was initially introduced for clinical application in 1968 in combination with sulfamethoxazole, also an inhibitor of folate biosynthesis, and later used alone to treat various infections [3,4]. TMP is widely prescribed to adults and to children, as well as to animals, worldwide [5,6,7]. In 2017, increasing concern over antibiotic resistance prompted the WHO to issue recommendations that include the reduction of TMP usage with food-producing animals as well as a complete restriction of its use with animals to promote growth and for preventive measures [8]. The goal of these recommendations is to lower the prevalence of antimicrobial resistance bacteria that could be transmitted to humans.

Multiple TMP resistance mechanisms have been reported [3]. The main mechanisms are the acquisition of type A (DfrA) or type B (DfrB) TMP-resistant dihydrofolate reductases. These TMP-resistant enzymes are expressed in addition to the TMP-sensitive, chromosomal FolA, allowing folate synthesis and bacterial survival. The DfrA family is homologous to FolA. It includes nearly 40 members that are 150–190 amino acids in length, similar to the FolA family [9]. The DfrB family currently consists of eight members; they are homotetrameric enzymes of 78 amino acids per protomer [10]. Contrary to DfrAs, they are phylogenetically and structurally unrelated to FolA [10,11]. Their evolutionary origin is currently unknown. DfrBs maintain full activity at the clinical concentrations of TMP that fully inhibit FolA [12]. They offer TMP resistance at concentrations at least 3 orders of magnitude greater than DfrAs [13], thus conferring TMP resistance that cannot be countered by administering TMP at higher concentrations.

DfrB1 is the first member of the DfrB family to have been reported. Over the past decades, DfrB1 has been characterized in great detail for its unique structure, its biophysical characteristics, its multimerization and its robustness [14,15,16,17,18]. In particular, DfrBs are distinguished from most enzymes in the fact that their single, central active site requires distinct contribution from each of the four identical protomers, creating an evolutionary conundrum [10,17]. Recently, other members of the DfrB family have been functionally characterized by our group, showing nearly indistinguishable dihydrofolate reductase activity, high resistance to TMP and similar inhibition by recently-reported inhibitors of distinct classes [19,20,21,22]. Nonetheless, at the outset of this study, none of the widely-used databases (CARD, ARDB and ARG-ANNOT) contained all eight known *dfrbs* sequences [23,24,25].

The prevalence of *dfrb* genes in clinical and environmental samples is currently unknown. The *dfrb* genes have rarely been reported in clinical samples [26,27]. Although this could be interpreted as the scarce presence of DfrBs in the collection of resistance genes in bacteria (resistome), it is important to note that the short 237-bp *dfrb* genes have not been routinely searched for. Until recently, gene-prediction algorithms used 300 nt as a cut-off to differentiate short non-protein-coding RNAs (ncRNA) from messenger RNAs (mRNA) [28]. Even now, sophisticated gene prediction algorithms such as the widely used Prodigal are unlikely to predict *dfrbs* as a result of their unusual codon usage and small size, both of which are penalized [29].

Experimental detection of TMP-resistant DfrB enzymes has also consistently failed to detect *dfrb* genes because of the prevalence of PCR-based methods: primers specific to *dfras* are used, with few or no primers specific to the unrelated *dfrbs* [30,31]. Fortunately, the advent of whole-genome sequencing now allows for large-scale computational screening of antimicrobial resistance gene databases beyond experimental biases.

According to Martínez et al., the greatest public health risk is observed when resistance genes to widely-used antibiotics are found on mobile genetic elements (MGE) of a human pathogen [32]. To date, a limited number of reports have found *dfrbs* near integrases and transposases, indicative of their genomic mobility [27,33]. Nonetheless, as *dfrbs* are rarely reported, their genomic context is essentially unexplored and the current public health risk that they represent is unknown.

Here, we searched publicly available databases to identify sequences containing *dfrbs*. We investigated the predicted pathogenicity of the organisms harbouring each sequence as well as the genomic context of *dfrbs*. We found that 70% of sequences containing *dfrbs* harbor virulence genes, mostly from *Pseudomonas*, a major cause of infection in humans that is difficult to treat because of its evolved resistance [34]. Overall, 97% of *dfrbs* are in proximity to a mobile genetic element, favoring their dissemination. Importantly, this investigation resulted in the identification of two new members of the *dfrb* family; we expressed both and confirmed their highly TMP-resistant phenotype in vitro. Our results demonstrate that the intrinsically TMP-resistant DfrBs can be found in a variety of contexts that are consistent with the transmission of multidrug resistance and justify closer surveillance of these genes.

## 2. Results

### 2.1. Expansion of the DfrB Family

Our first objective was to determine whether further DfrB homologues could be identified, to join the small but rapidly growing DfrB family. Using profile hidden Markov models (HMM) of the six functionally characterized DfrBs (DfrB1–5 and DfrB7) [19], we searched the TrEMBL database. In addition to confirming the presence of DfrB1–9, we also identified two genes displaying high homology to the conserved core of DfrBs yet sharing sequence identity of less than 95% to any known *dfrb* were identified. They were named *dfrb10* and *dfrb11* (Table 1).

DfrB10 was found on the p12969-DIM mega-plasmid (0.4 Mb) from a *Pseudomonas putida* strain isolated in China in 2013 from a patient suffering from pneumonia [35]. DfrB11 was identified in a groundwater sample at the Horonobe Underground Research laboratory in Japan in 2017, in a *Betaproteobacteria* sequence [36]. Both new DfrBs procure the same phenotype as the other DfrBs when overexpressed in *E. coli*: they confer resistance to 0.6 mg/mL TMP, the highest concentration of TMP that can be solubilized in 5% methanol (Table 1). This situates these genes amongst the most resistant dihydrofolate reductases known to date.

### 2.2. Identification of Bacterial Sequences

The DNA sequences of the eight previously reported *dfrbs* and the two new *dfrbs* were searched against publicly available genomic databases; we note that there is no DfrB8 (Appendix A
Appendix A) [11]. Since our objective was to analyze the genomic context of *dfrbs*, we retained only genomic segments that include at least 10kb both upstream and downstream from a *dfrb*. A total of 110 sequences were collected, representing 16 different bacterial species. In some cases, multiple similar sequences originated from the same BioProject; in these instances, redundant sequences were excluded from further analysis, keeping one representative sequence.

The taxonomic summary for the 61 remaining sequences is presented in Table 2. All sequences but one came from *Gammaproteobacteria* and included three different orders: *Aeromonadales, Enterobacterales* and *Pseudomonadales*. The predominant species was the clinically-relevant *Pseudomonas aeruginosa*, accounting for 36 sequences (59%). The only *Betaproteobacteria* sequence was from *Burkholderia dolosa*, isolated from a cystic fibrosis patient in the United States of America [37].

When available, information on the isolation source of each sample and the country of origin was compiled. Most *dfrb*-containing strains were identified in samples collected in Asia (39%), followed by Europe (20%), America (17%) and Africa (5%). The majority of strains (62%) were found within humans; an additional 7% found in wastewater and 2% in hospital wastewater may also be of human origin. Surprisingly, despite intensive use of TMP for livestock, only 3% of strains from our dataset were isolated from animals. This could indicate a sampling bias from the databases. Finally, 2% of samples were identified as environmental.

Four species in the dataset (*Acinetobacter baumannii*, *Enterobacter hormaechei*, *Klebsiella pneumoniae* and *Pseudomonas aeruginosa*) are categorized as ESKAPE pathogens, accounting for 67% of the sequences. Overall, 79% of the 61 *dfrb*-containing sequences contain virulence genes enabling them to cause infection according to the VFDB database. Because our genomic sequence dataset included partial sequences, the fraction of sequences of pathogenic bacteria can be underestimated if virulence genes are outside of the sequenced region.

### 2.3. Analysis of the Genomic Context

We investigated the genomic context within which the *dfrbs* were found. In particular, the presence of MGEs can inform us of the capacity of the *dfrbs* to transfer to other genomes. Ever since the initial discovery of *dfrb1* (R67) and *dfrb2* (R388) on plasmids, DfrBs have been systematically referred to as being plasmid-borne [38,39]. This is consistent with the importance of plasmids in acquired bacterial resistance [40]. The genomic context was determined by classifying the *dfrb*-containing genomes as either plasmidic or chromosomal using the PlasFlow classification tool [41]. This resulted in 18 sequences being labelled “plasmids” (30%) and 43 sequences labelled “chromosomes” (70%) (Figure 1). The only sequence identified in *Betaproteobacteria* was chromosomal. Among the *Gammaproteobacteria*, all 16 sequences identified in *Enterobacterales* were plasmidic, whereas the two sequences in *Aeromonadales* were chromosomal. The *Pseudomonadales* sequences were chromosomal except for two sequences: one from *P. aeruginosa* and one from *P. putida*. Only one plasmidic sequence, that from *P. rettgeri*, was labelled as pathogenic.

Next, we gained insight into the types of genes flanking the *dfrbs*. MGEs and other resistance genes near the *dfrbs* would define them as belonging to a multiresistance context. A blastx search was performed, using 20kb sequence segments containing *dfrbs* as queries, against a compiled antibiotic resistance gene database (see Materials and Methods), keeping hits having at least 80% coverage and 60% identity. We first determined that the vast majority of *dfrbs* (89%) had both integrase and transposase within a 10kb window (Figure 1). Five sequences (NZ_CP010378.1, NZ_SWEG01000001.1, NZ_UWXD01000002.1, NZ_CP032569.1, NZ_KU130294.1) had only an integrase annotated nearby and one sequence (CP031876.1) had only a transposase annotated. Two sequences included neither; one (NZ_CM002277.1) was a chromosomal pathogenic sequence from *Burkholderia dolosa*, the only *Betaproteobacteria* identified. That sequence included no further genes related to antimicrobial resistance or genomic mobility within 10kb of its *dfrb* (Figure 2a). The other sequence including no integrase or transposase (AOBK03000081.1) was a chromosomal and pathogenic sequence from *P. aeruginosa*. It included only the rifampin-resistance gene *arr2* at a distance of 0.14kb from its *dfrb*.

The distance separating the *dfrbs* from the integrases and transposases was mapped (Figure 3). For *dfrb1*, *dfrb2* and *dfrb4*, we observed a large variability in those distances, suggesting diversity in their genomic context and thus diversity in the events of integration. In the cases of *dfrb3* and *dfrb5*, the same gene cassette was present in a few sequences, thus the same distance between elements was mapped. For example, all eight *dfrb3* genes were found in integrons directly upstream from the class 1 integrase, marking *dfrb3* as the gene most recently integrated into the cassette. Interestingly, *dfrb3* was found only in plasmids, in *Enterobacterales*. Six of these genes were found in the short *dfrb3*/*sul1* cassette. Similarly, multiple sequences held the same cassettes containing *dfrb5*. There were 15 sequences with the *aac(6′)-Il*/*vim-2*/*dfrb5*/*aac(3)-Id* cassette and five with the *aac(6′)-Il*/*dfrb5*/*aac(3)-Id* cassette; some other cassettes were found twice.

Finally, we mapped antimicrobial resistance genes annotated in a window 10kb on either side of the *dfrbs* and classified them according to the antimicrobial to which they conferred resistance (Figure 2a,b). The most prevalent phenotype was aminoglycoside resistance (80% of all sequences), followed by beta-lactamase genes (60%). Surprisingly, although TMP is often prescribed in combination with sulfonamide [42], only 57% of sequences included sulfonamide-resistance genes. Resistance to metals, phenicol, fluoroquinolones, rifampicin, macrolides, lincosamides, streptogramines, efflux genes and tetracyclin were observed at a lower frequency.

## 3. Discussion

The overwhelming majority of studies on clinical resistance to trimethoprim have focused exclusively on DfrAs. Recently, Sánchez-Osuna et al. reported two mechanisms of DfrA evolution [9]. One mechanism involves the mutation and mobilization of trimethoprim-sensitive FolA genes (ex. *A. baumannii*
*folA* mutated to *dfrA39* and *dfrA40*); the second relies on the mobilization of intrinsically trimethoprim-resistant *folA* genes. Trimethoprim resistance through DfrAs evolves readily, explaining the large number of DfrA genes: the recent addition of four members to the DfrA family has brought it to nearly 40 members [9].

Unlike DfrAs, the evolutionary origin of DfrBs has not been investigated. Little is known about these peculiar homotetrameric enzymes, including their prevalence and emergence in clinical and environmental samples [27]. We examine, for the first time, the genomic context of *dfrbs* by analyzing publicly available sequences containing *dfrbs*. By reporting the microorganisms that harbor them and determining whether they occur in the context of genetic mobility and/or resistance to multiple antibiotics, we provide insight into the risk they represent for public health.

Using a query-set consisting of the eight previously known *dfrb* genes and two newly identified and confirmed TMP-resistant *dfrb* genes, we identified 61 different genomic sequences containing *dfrbs*. The country of origin of each sample illustrates that *dfrbs* are dispersed worldwide. The vast majority of *dfrbs* (74%) for which the source of isolation is known are related to human activities, whereas only one sequence comes from an environmental sample and 25% are from unknown sources. This observation could well reflect sampling biases due to overrepresentation of studies related to human activities relative to environmental studies in genomic databases. In Canada, TMP and sulfonamides are the fourth most highly prescribed antimicrobials for animals, representing 57,865 kg in 2018 [43], justifying the importance of increasing genomic analyses of animal samples to determine the prevalence of *dfrbs* in all relevant contexts.

Amongst the chromosomal sequences we identified, 93% contained virulence genes and included at least one MGE near the *dfrb* gene. These combined criteria define the highest risk that antimicrobial resistance genes can present [32]. The remaining sequences containing *dfrbs* included at least one of these two criteria. All plasmid-borne *dfrbs* were near an MGE, allowing them to spread easily among bacteria.

Since their discovery, DfrBs have been considered to be solely plasmid-borne [44,45]. Indeed, *dfrb1* was first observed in an *E. coli* strain where it is plasmid-borne, leading to the incorrect assumption that *dfrbs* are always plasmidic [38]. Nevertheless, we observed not only a few exceptions to this long-standing conjecture, but rather that only 29% of *dfrb*-containing sequences in our sample were plasmidic. The *Enterbacterales* species harbor *dfrbs* on plasmids, while *Aeromonadales* and *Pseudomondales* species harbor *dfrbs* on their chromosome (Figure 1).

It is interesting to note that the only two *dfrb* genes that were not found near MGE were in chromosomal sequences (Figure 1). This suggests that *dfrbs* might have mobilized from a chromosome to a plasmid, and not have originated from plasmids. Further thorough examination of *dfrbs* and their mobilization context, both in plasmids and chromosomes, would allow the retracing of the early events of *dfrb* mobilization.

DfrB1 and DfrB2 were discovered in the 1970s, subsequent to the introduction of TMP. DfrB1, also named R67, dfrII and dfr2a, has been extensively characterized for its structure, assembly and catalytic activity [14,16,17,46,47,48,49]. Only recently have other members of the family been characterized [19] or even reported, such that it could have been thought that DfrB1 is the most widespread among DfrBs. However, *dfrb1* was identified in only 16% of the 61 sequences identified here. Unexpectedly, *dfrb5* was identified in 38% of the sequences, in various genomic contexts and geographical locations, suggesting that is more broadly disseminated than *dfrb1*. In addition, 23% of sequences contained *dfrb2*, 13% contained *dfrb3* while 8% contained *dfrb4*. The *dfrb10* gene, reported here for the first time, was identified in one sequence and none among the *dfrb6*, *dfrb7*, *dfrb9* or *dfrb11* genes were identified using our search criteria for genomic segments.

Although the sample sets included in the databases we searched represent only a fragmentary picture of gene dissemination, some members of the DfrB family are clearly more prevalent while others may not have yet emerged. Interestingly, the most prevalent *dfrb5* is closely related to the first-reported *dfrb1* (Appendix A). Further investigation will be required to determine whether early events of *dfrb* mobilization in pathogenic bacteria involves either of these two genes. Close surveillance of the prevalence of these genes is needed to evaluate the spread of this family of genes.

All but one *dfrb* (NZ_CM002277.1) were found near at least one antibiotic resistance gene (ARG), the majority (83%) being in proximity (less than 10kb) to three other ARGs. Among these, the *sul1* sulfonamide-resistant gene is present in the vast majority of clinically relevant integrons [50]. We previously reported identification of *dfrb4* in a clinical class 1 integron within the *dfrb4*/*qacEΔ1*/*sul1* cassette, flanked by further resistance genes [27]. Since TMP is often prescribed in combination with sulfamethoxazole, it is noteworthy that only 57% of the *dfrbs* identified in integrons in this study were colocalized with *sul1*. The *dfrbs* were found in class 1 integrons as defined by the presence of a class 1 integrase. Multiple ARGs were observed within the same cassettes as the *dfrbs* (Figure 2a). Expression of some of these ARGs (e.g., ß-lactam resistance *vim-2* and *oxa-10*, aminoglycoside resistance *aadA1*, rifampin resistance *arr-2*) in other class 1 integrons has previously been reported [51,52,53]. Although this does not demonstrate gene expression in these genomic contexts, it is consistent with the hypothesis that the ARGs as well as the *dfrbs* are expressed in the class 1 integrons identified here.

No *dfras* were found in proximity to the *dfrbs*; this is expected since these genes procure the same phenotype. Nonetheless, duplication of *dfrb5* was observed in the integron of one genome (NZ_CP031449.2), where three copies of *dfrb5* were observed in the same integron (Figure 2a). In addition, duplication of similar integrons containing *dfrb5* in the same sequence was observed in one plasmid and four genomes, all from *P. aeruginosa*. Considering the incomplete nature of the sequences we analyzed, it is possible that a greater number of amplification events could be identified upon analysis of longer sequence segments.

Given the importance of TMP both in the clinic and with livestock, it is critical to monitor the emergence of resistance to this antimicrobial. TMP resistance has generally been associated with DfrAs. Here, we have demonstrated that monitoring the emergence and prevalence of DfrBs will provide important insights into global TMP resistance and thus contribute to policy making to contain the spread of antimicrobial resistance.

## 4. Materials and Methods

### 4.1. Identification of Putative Type B Dihydrofolate Reductases

The six DfrB sequences that were previously functionally characterized (DfrB1–DfrB5 and DfrB7) [19] were used to create a profile hidden Markov models (HMM) with HMMER version 3.3 (http://hmmer.org/ accessed on 10 January 2021). This profile was used as a query against the UniProtKB/TrEMBL database (22 April 2020 release, 184,998,855 sequences) [54]. Hits with E-value lower than 1 × 10^−40^ were considered and compared to known DfrB sequences. Predicted sequences having a protein sequence identity lower than 95% relative to any known DfrB sequence were considered as new genes.

### 4.2. Subcloning of dfrb10 and dfrb11

The genetic sequences of *dfrb10* and *dfrb11* were obtained in pUC57 (BioBasic) according to the Genbank accession numbers in Table 1. The N-terminally His_6_-tagged ORF sequences of *dfrb10* and *dfrb11* were subcloned into pET24 (Qiagen) downstream of the lactose operon repressor, following the lac operator sequence, using the *Nde*I and *Hind*III restriction sites. Both genes were amplified by PCR using the same forward primer 5′-GAAATAATTTTGTTTAACTTTAAGAAGGAGATATA**CATATG**AGAGGATCTCACCATCAC-3′ (*Nde*I site in bold) and a reverse primer that differs at one base (underlined) to maintain the native stop codon, *dfrb10*: 5′-GGTGGTGCTCGAGTGCGGCCGC**AAGCTT**TTAGGCCACGCG-3′; *dfrb11*: 5′-GGTGGTGCTCGAGTGCGGCCGC**AAGCTT**TCAGGCCACGCG-3′ (*Hind*III site in bold). Phusion HF polymerase (ThermoFisher) was used according to the manufacturer’s protocol, using 55 °C as the annealing temperature. Amplified genes, as well as pET24, were digested with *Hind*III (NEB) for 14 h and *Nde*I (NEB) for 2 h at 37 °C, followed by enzyme inactivation for 20 min at 80 °C. They were gel-extracted using the Monarch DNA gel extraction kit (NEB) and purified using the DNA Cleanup kit (NEB). Inserts were ligated into digested pET24 using a DNA ligation kit (Takara) according to the manufacturer’s instructions. Briefly, the digested gene and pET24 vector were incubated at 16 °C for 3 h in Takara solution I. The ligation products were transformed into CaCl_2_-competent *E. coli* DH5α prepared by the method of Inoue [55]. The DNA sequences of *dfrB10*-pET24 and *dfrB11*-pET24 were confirmed by DNA Sanger sequencing (Genome Quebec platform at Sainte-Justine Hospital). The final constructs yield N-terminally, His_6_-tagged DfrB proteins. His_6_-DfrB3 in pET24 was previously reported [19]. The negative control cTEM-19m, with an expressible β-lactamase insert instead of a *dfrb* insert, was previously described [56].

### 4.3. Minimal Inhibitory Concentration

MICs were determined according to Wiegand et al. [57] using the broth microdilution method. Briefly, *E. coli* BL21(DE3) cells expressing His_6_-DfrB3 (positive control), His_6_-DfrB10, His_6_-DfrB11 or cTEM-19m (negative control) were propagated overnight in Luria-Bertani (LB) media with 50 µg/mL kanamycin. In 96-well plates, an inoculum of 10^5^ colony forming units (cfu) was inoculated in LB media, with 0.1 mM IPTG (ThermoFisher) and TMP (Sigma) in 2-fold concentration steps up to 600 µg/mL; the latter is the highest concentration of TMP soluble in 5% methanol. MICs were determined in triplicate.

### 4.4. Download of Genomes

The sequences of *dfrb1*-*dfrb7*, *dfrb9* and the newly identified *dfrb10* and *dfrb11* were used as queries for blastn 2.10.0 searches against four genomic databases (performed on the 2020.07.04): RefSeq (bacterial sequences), GenBank (bacterial sequences) and the Microbial Complete Genomes and Complete Plasmids databases found at https://blast.ncbi.nlm.nih.gov/Blast.cgi (accessed on 10 January 2021) [58,59,60]. Genomes containing at least one query sequence and having a sequence length of at least 10kb both upstream and downstream of the *dfrb* sequences were collated. In total, 110 sequences were identified and served for analysis.

### 4.5. Protein Database Constructions

The following protein databases were downloaded on 2020.07.07: Integrase, IntI1 and sul1 databases from the I-VIP pipeline [61], ARG-ANNOT [25], ICEberg 2.0 [62], CARD [23], BacMet [63] and UniProtKB/Swiss-Prot [64]. These databases were merged and redundant sequences were removed. Two genes, coding for 78 and 97 amino-acid products, were both named DfrB1; the shorter gene version was kept to match the consensus length of all other members of the family.

### 4.6. Annotation

The 110 sequences of 20 kb (10 kb upstream and downstream of a *dfrb*) were used as query sequences against the blast database for a blastx 2.10.0 search with the parameters of E-value lower than 1 × 10^−10^ and culling_limit of 1. Hits with coverage of ≥80% and protein identity of ≥60% were kept. Where multiple sequences from a same NCBI BioProject presented the same annotation, all but one were removed from the dataset. In total, 61 sequences served for analysis.

### 4.7. Classification of Sequences as Chromosomal or Plasmidic

The 110 sequences were classified as chromosomal or plasmidic using PlasFlow 1.1 with a threshold of 0.65 [41].

### 4.8. Identification of Pathogenic Hosts

For each sequence, a blastx 2.10.0 analysis was carried against virulence factor protein sequences from the core dataset of VFDB (last update on 17 July 2020) [65] using an e-value cutoff of 1 × 10^−15^. Hits were filtered using an identity and coverage threshold of 60%. Sequences with one or more hit were labelled pathogenic.

### 4.9. Phylogenetic Tree

The phylogenetic tree was constructed with NGPhylogeny.fr [66] using the host species’ 16S rRNA from the NCBI reference genome. The tools MUSCLE (with the Neighbor joining option), Noisy, PHyML + SMS, and Newick were used for tree construction. The tree was annotated with iTOL, where the GenBank accession of each sequence is displayed [67].

## Figures and Tables

**Figure 1 antibiotics-10-00433-f001:**
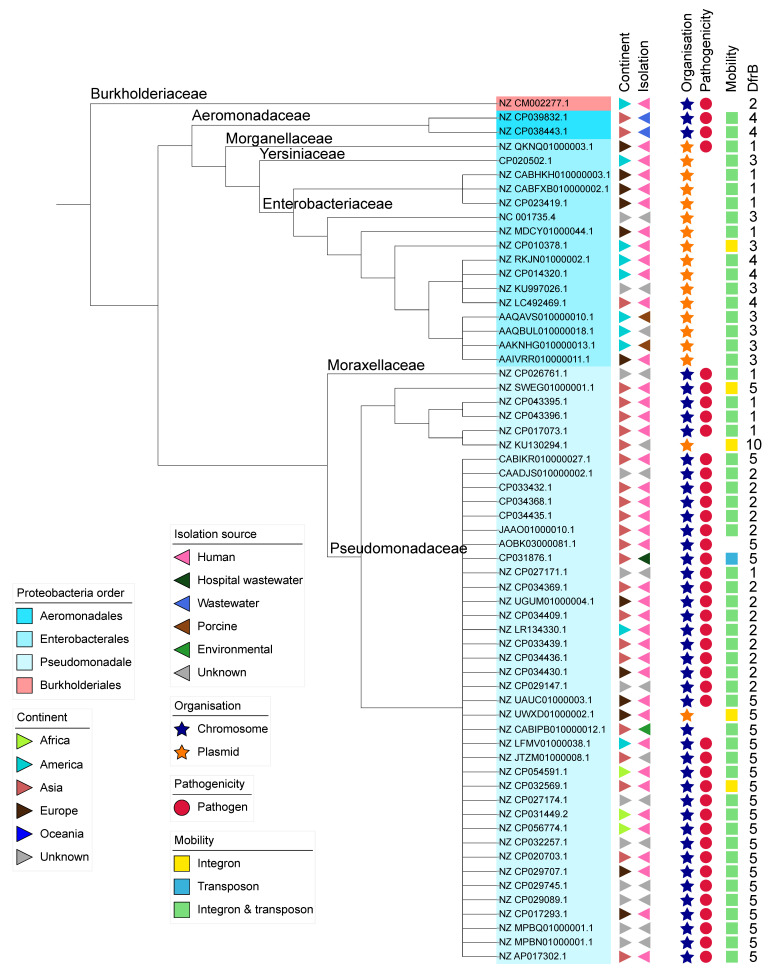
Annotated phylogenetic tree of species harboring a *dfrb*. Taxonomic classification of order and family is followed by categorization according to GenBank information on the strain’s isolation source and country of isolation. Sequences are further categorized as being located in a chromosome or a plasmid, pathogenicity of the host organism and information on mobile genetic elements. The *dfrb* gene member identified in each sequence is specified (i.e., “2” indicates *dfrb2*).

**Figure 2 antibiotics-10-00433-f002:**
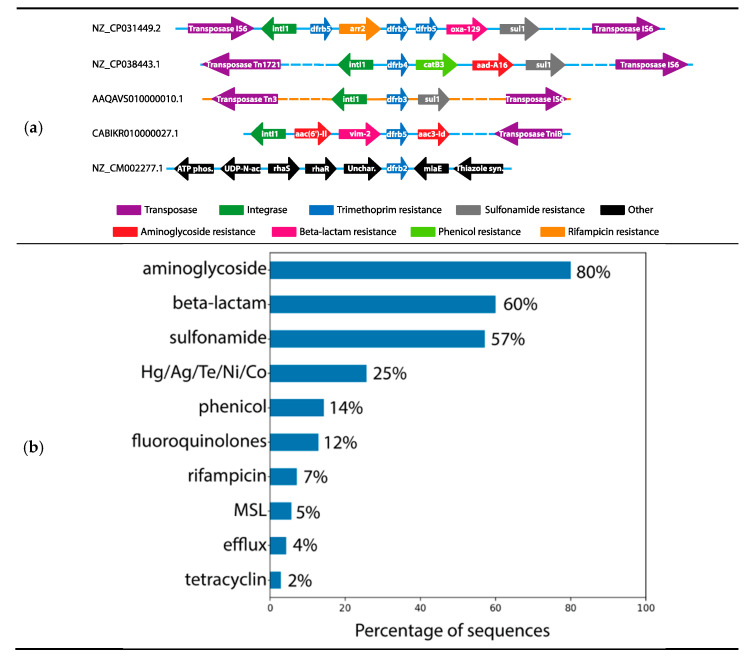
(**a**) Scheme of the genomic context of five representative sequences. Blue lines represent genomic sequences, whereas the orange line represents a plasmid sequence. Dashed lines contain regions that are not represented here. (**b**) Population of *dfrb* genes accompanied by a gene conferring resistance to another antimicrobial agent, expressed as percentage. MSL: macrolides, streptogramins, lincosamides.

**Figure 3 antibiotics-10-00433-f003:**
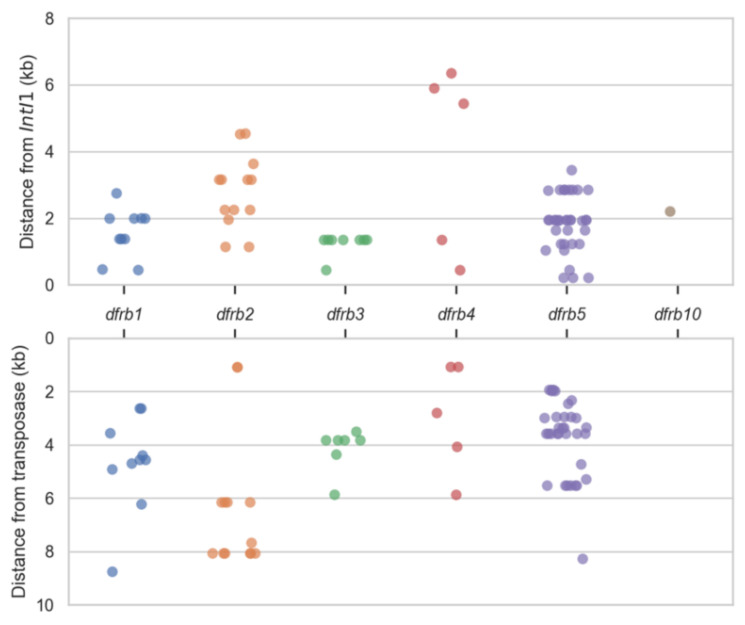
Distance between *dfrbs* and genes associated with genomic mobility. Top panel: distance between the *dfrbs* that are downstream a class 1 integrase. Bottom panel: distance between *dfrbs* and the closest transposase. Each dot represents one *dfrb* gene.

**Table 1 antibiotics-10-00433-t001:** Information and MICs on the newly identified DfrB10 and DfrB11.

New Name	UniprotKB Accession Number	Genbank Accession Number	Closest Characterized DfrB (Protein Identity/ DNA Identity) ^a^	MIC (µg/mL)
DfrB10	A0A2Z1CLP9	ALZ46148.1	DfrB3 (92%/93%)	>600
DfrB11	A0A2N2TNN4	PKO69073.1	DfrB3 (90%/87%)	>600

^a^ Protein sequence identity of all members of the DfrB family are reported in Appendix A
Appendix A.

**Table 2 antibiotics-10-00433-t002:** Taxonomic classification of all strains identified that include at least one *dfrb.*

Class/Order/Family/Genus	Strain Count ^a^
**Betaproteobacteria**	1
***Burkholderiales***	1
*Burkholderiaceae*	1
*Burkholderia*	1
**Gammaproteobacteria**	60 (110)
***Aeromonadales***	2
*Aeromonadaceae*	2
*Aeromonas*	2
***Enterobacterales***	16 (17)
*Enterobacteriaceae*	14 (15)
*Citrobacter*	1
*Enterobacter*	1
*Escherichia*	4
*Klebsiella*	4 (5)
*Salmonella*	4
*Morganellaceae*	1
*Providencia*	1
*Yersiniaceae*	1
*Serratia*	1
***Pseudomonadales***	42 (91)
*Moraxellaceae*	1
*Acinetobacter*	1
*Pseudomonadaceae*	41 (90)
*Pseudomonas*	41 (90)

^a^ Values include sequences used in the analysis after exclusion of redundancy. Values in parentheses include redundant sequences.

## Data Availability

Data is contained within the article or supplementary material. The accession numbers of analyzed sequences listed in Figure 1 are accessible in publicly available databases.

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
