# Peer review of "The Bacterial Genomic Context of Highly Trimethoprim-Resistant DfrB Dihydrofolate Reductases Highlights an Emerging Threat to Public Health"

_antibiotics, 2021, doi:10.3390/antibiotics10040433_

Round 1

Reviewer 1 Report

I would suggest adding your dfrB10 and dfrB11 to Figure 3 for reference.

Did you test the ability of your HMM profile to identify known examples of dfrb6, dfrb7,and dfrb9?

Line 374, should this also be 116 sequences also, not 110?

Author Response

  1. I would suggest adding your dfrB10 and dfrB11 to Figure 3 for reference.

Response: This is an interesting suggestion. DfrB10 is present once in Fig. 3, having been identified in a genomic context that satisfied our minimal criteria for analysis. Specifically, Fig 3 represents the genomic segments where the dfrb gene was flanked with a minimum of 10 kb of genetic material both up and downstream. That requirement allowed controlled assessment of mobile genetic elements / other genes of interest. The database search did not yield a dfrB11 gene in a genomic context that satisfied those minimal criteria for analysis. The same applied to dfrb6, dfrb7 and dfrb9. To ensure coherence of the data analysis, we prefer not to include the short genomic segment that allowed identification of the dfrb11 gene.   

  1. Did you test the ability of your HMM profile to identify known examples of dfrb6, dfrb7,and dfrb9?

Response: Yes, we have. We now specify this, and also clarify the process of identification of the new dfrb genes, as follows: (section 2.1 first paragraph) “Using profile hidden Markov models (HMM) of the six functionally characterized DfrBs (DfrB1-5 and DfrB7) [19], we searched the TrEMBL database. In addition to confirming the presence of DfrB1-9, we also identified two genes displaying high homology…”

  1. Line 374, should this also be 116 sequences also, not 110?

Response: Thank you for noticing; in fact the ‘116’ in the first line of section 4.6 has been corrected to ‘110’.

Reviewer 2 Report

The article of Claudèle Lemay-St-Denis et al. reports a genetic and genomic analysis, looking for the dfrb genes and related upstream and downstream sequences, for about 10 kbp (20 kbp in total). Upstream and downstream sequences showed ORFs that can potentially encode virulence factors. The authors also report the bacterial taxa in whose genomes or plasmids these sequences were found.

I believe that some points of the article (listed below as a major revision) need to be corrected appropriately. The manuscript is written quite well. Detected results implement the information on bacterial dfrb gene although, in terms of originality, I don’t find it particularly appropriate for a journal having IF 3.893.

Major revision

Abstract

Line 20. Please, change: .. those sequences harbor virulence factors..  in 

..sequences harbor also virulence genes ...

The authors mapped/detected genes, and not the products. Alias virulence factors.

Line 21. Change: .. representing the highest risk for antibiotic resistance genes.  In 

 .. representing a potential risk for antibiotic resistance genes.

The authors should demonstrate experimentally the highest risk !

At the moment the authors mapped only sequences. Do they encode these genes for virulence factors ? The presence of a gene/ORF does not mean that it encode a product in any context or environment. It should be demonstrated, at least in the case of human or animals. Are detected promoters or further regulatory sequences ? If there are genes (among the 61 sequences here analyzed) for which there are studies, please add references and discuss. If they are not there, it is better to be cautious. For instance, in the discussion the authors cite one gene, sul1. Is the sul1 in the same genomic/sequence context found ?

DNA regulatory sequences are fundamental for the gene expression. Why didn’t the authors searched for them ? At least their presence may be a really signal of potential risk.

Introduction

Lines 93-94-95. The authors wrote: “Our results demonstrate that the intrinsically TMP-resistant DfrBs are a threat to public health and justify closer surveillance of these genes.”

Please correct this sentence.

Really, the author’s findings does not demonstrate a threat to public health. It was already demonstrated. The Author’s findings show two variant (dfrb 10 and 11) and 61 sequences including the dfrb gene. The authors also map putative genes, upstream and downstream the dfrb gene.

Material and Methods

Line 318. Section 4.2 Subcloning of dfrb10 and dfrb11.

What does it mean, Dfrb10 and dfrb11 were cloned into (line 320) … , Both genes were amplified (line 321) ….

The authors must briefly specify/describe what they really cloned: the ORFs, the ORFs plus the Dfrb10 and dfrb11 original Ribosomal Binding Sequences, etc.

A reader should read papers, and  the manual of pET24 to understand. Material and Methods is a fundamental section of an article, and a briefly description of all experimentally aspects must be reported.

Minor revision

Title

Line 2. Add bacterial in the title, (e.g. The bacterial genomic context….).

Abstract

Line 26. Change: .. TMP-resistant dfrbs are an emerging threat to public health … in   

… TMP-resistant dfrbs are a potential emerging threat to public health

Introduction

Line 89. Change: Virulence factors         in       virulence genes

Matherial and Methods

Line 331. Please, write HMM and HMMR in full, at least the first time. Even in the results or discussion. Please, in general, write in full all acronyms, when occur for the first time.

Line 321. Please, add the plasmid (pET24) company.

Linee 324-325. Specify that the two reverse primers differ for one base. It is fast to follow.

Line 330-331. Please describe briefly the ligation. If for any approach is necessary read a reference, the reading of the article will be long.

Line 332. Please specify Sanger sequencing.

Discussion

line 255 Change: Virulence factors         in       virulence genes

Author Response

General response: Thank you for your thorough and expert revision of the manuscript. We appreciate the requested changes provide a significant improvement.

Abstract

  1. Line 20.Please, change: .. those sequences harbor virulencefactors..  in ..sequences harbor also virulence genes ...

The authors mapped/detected genes, and not the products. Alias virulence factors.

Response: We have corrected this instance and other instances of this error.

  1. Line 21.Change: .. representing thehighest risk for antibiotic resistance genes.  In 

 .. representing a potential risk for antibiotic resistance genes.

The authors should demonstrate experimentally the highest risk !

Response: Indeed, in the context of the abstract, ‘potential risk’ is clearer.

  1. At the moment theauthors mapped only sequences. Do they encode these genes for virulence factors ? The presence of a gene/ORF does not mean that it encode a product in any context or environment. It should be demonstrated, at least in the case of human or animals. Are detected promoters or further regulatory sequences ? If there are genes (among the 61 sequences here analyzed) for which there are studies, please add references and discuss. If they are not there, it is better to be cautious. For instance, in the discussion the authors cite one gene,sul1. Is the sul1 in the same genomic/sequence context found ?

DNA regulatory sequences are fundamental for the gene expression. Why didn’t the authors searched for them ? At least their presence may be a really signal of potential risk.

Response: The reviewer is correct that we have not demonstrated expression of DfrBs in the genomic contexts identified here. That is a longer-term objective of this work and will include a thorough characterization of the DNA regulatory sequences. Indeed, there are relatively few examples where expression of antibiotic resistance genes (ARG) is demonstrated in their native context. To clarify what has been demonstrated here, we now refer to demonstrated expression for examples among the ARG identified here and better define the genomic context that relates them, as follows: “Among these, the sul1 sulfonamide-resistant gene is present in the vast majority of clinically relevant integrons [50]. We previously reported identification of dfrb4 in a clinical class 1 integron within the dfrb4/qacE∆1/sul1 cassette, flanked by further resistance genes [27]. Since TMP is often prescribed in combination with sulfamethoxazole, it is noteworthy that only 57% of the dfrbs identified in integrons in this study were colocalized with sul1. The dfrbs were found in class 1 integrons as defined by the presence of a class 1 integrase. Multiple ARG were observed within the same cassettes as the dfrbs (Fig. 2a). Expression of some of these ARG (e.g. ß-lactam resistance vim-2 and oxa-10, aminogly-coside resistance aadA1, rifampin resistance arr-2) in other class 1 integrons has previously been reported [51–53]. Although this does not demonstrate gene expression in these genomic contexts, it is consistent with the hypothesis that the ARG as well as the dfrbs are expressed in the class 1 integrons identified here.”

Introduction

4. Lines 93-94-95. The authors wrote: “Our results demonstrate that the intrinsically TMP-resistant DfrBs are a threat to public health and justify closer surveillance of these genes.”

Please correct this sentence.

Really, the author’s findings does not demonstrate a threat to public health. It was already demonstrated. The Author’s findings show two variant (dfrb 10 and 11) and 61 sequences including the dfrb gene. The authors also map putative genes, upstream and downstream the dfrb gene.

Response: We have modulated that sentence as follows: “Our results demonstrate that the intrinsically TMP-resistant DfrBs can be found in a variety of contexts that are consistent with transmission of multidrug resistance and justify closer surveillance of these genes.”

Material and Methods

5. Line 318.Section 4.2 Subcloning of dfrb10 and dfrb11.

What does it mean, Dfrb10 and dfrb11 were cloned into (line 320) … , Both genes were amplified (line 321) ….

The authors must briefly specify/describe what they really cloned: the ORFs, the ORFs plus the Dfrb10 and dfrb11 original Ribosomal Binding Sequences, etc.

A reader should read papers, and  the manual of pET24 to understand. Material and Methods is a fundamental section of an article, and a briefly description of all experimentally aspects must be reported.

Response: We have clarified the subcloning section of our methods, as follows. “The N-terminally His6-tagged ORF sequences of dfrb10 and dfrb11 were subcloned into pET24 (Qiagen) upstream of the lactose operon repressor, following the lac operator sequence, using the NdeI and HindIII restriction sites. Both genes were amplified by PCR using the same forward primer 5'-GAAATAATTTTGTTTAACTTTAAGAAGGAGATATACATATGAGAGGATCTCACCATCAC-3' (NdeI site in bold) and a reverse primer that differs at one base (underlined) to maintain the native stop codon, dfrb10: 5'-GGTGGTGCTCGAGTGCGGCCGCAAGCTTTTAGGCCACGCG-3'; dfrb11: 5'-GGTGGTGCTCGAGTGCGGCCGCAAGCTTTCAGGCCACGCG-3' (HindIII site in bold).”

Minor revision

Title

6. Line 2. Add bacterial in the title, (e.g. Thebacterial genomic context….).

Response: Done.

Abstract

7. Line 26. Change: .. TMP-resistantdfrbs are an emerging threat to public health … in   

… TMP-resistant dfrbs are a potential emerging threat to public health

Response: Done.

Introduction

8. Line 89. Change: Virulencefactors         in       virulence genes

Response: Done.

Material and Methods

9. Line 331.Please, write HMM and HMMR in full, at least the first time. Even in the results or discussion. Please, in general, write in full all acronyms, when occur for the first time.

Response: HMM - Done. HMMER is the name of the software package for which there is a link to access. We have verified the manuscript for all the other non-standard acronyms.

10. Line 321. Please, add the plasmid (pET24) company.

Response: Done.

11. Linee 324-325. Specify that the two reverse primers differ for one base. It is fast to follow.

Response: Done. The differing base is now underlined and the purpose of that difference is now stated.

12. Line 330-331. Please describe briefly the ligation. If for any approach is necessary read a reference, the reading of the article will be long.

Response: Done, as follows: “Inserts were ligated into digested pET24 using a DNA ligation kit (Takara) according to the manufacturer’s instructions. Briefly, the digested gene and pET24 vector were incubated at 16ËšC for 3h in Takara solution I. The ligation products were transformed into CaCl2-competent E. coli DH5a prepared by the method of Inoue [55]. »

13. Line 332.Please specify Sanger sequencing.

Response: Done.

14. Discussion

line 255 Change: Virulence factors         in       virulence genes

Response: Done.

Reviewer 3 Report

The authors are presenting an analysis of DfrB genes mediating resistance to trimethoprim. The study includes the identification of novel dfrb  genes as well as the prevalence of the DfrB family of genes across multiple bacterial families. The manuscript is well-written and easy to read. The significance of the study is that it expands our understanding of the distribution of dfrb genes and their proclivity for mobility, exacerbating the antibiotic resistance crisis.

I find no major flaws or concerns with the manuscript.

Minor points:
1. Line 89-90: misspelled "humans"
2. 306: is there a word missing in this phrase "...will allow to gain important..." ?
3. Table 2: it is suggested that the authors employ a uniform vertical alignment for the strain count numbers rather than the present staggered alignment.

Author Response

  1. Line 89-90: misspelled "humans"

Response: Done.

2. 306: is there a word missing in this phrase "...will allow to gain important..." ?

Response: Thank you, we revised the sentence which now reads “Here, we have demonstrated that monitoring the emergence and prevalence of DfrBs will provide important insights into global TMP resistance…. »

3. Table 2: it is suggested that the authors employ a uniform vertical alignment for the strain count numbers rather than the present staggered alignment.

Response: Done.

General comments

In addition, we have noted a mistake in the sequence id and gene orientation of Fig2a, which has been changed. We also corrected a mistake in percentage of sequences where dfrb5 was identified at the end of page 8. We also changed figure numbering to be consistent with their appearance in the text.

Round 2

Reviewer 2 Report

The article by Claudèle Lemay-St-Denis et al. has been properly revised. Provided corrections are now more “sound” with the results.

I advise authors to re-read the text and change some terms with technically more appropriate words.

Just for instance:

Line 109. Expansion..!         Better spread.. or  propagation...

After this the article could be considered for publication.